# The Validity of Functional Threshold Power and Maximal Oxygen Uptake for Cycling Performance in Moderately Trained Cyclists

**DOI:** 10.3390/sports7100217

**Published:** 2019-10-01

**Authors:** Arne Sørensen, Tore Kristian Aune, Vegar Rangul, Terje Dalen

**Affiliations:** 1Department of Physical Education and Sport Science, Faculty of Education and Arts, Nord University, 7600 Levanger, Norway; tore.k.aune@nord.no (T.K.A.); terje.dalen@nord.no (T.D.); 2Faculty of Health Science, Nord University, 7600 Levanger, Norway; vegar.rangul@nord.no

**Keywords:** 20 min FTP, VO_2max_, mass-start mountain bike race, validity

## Abstract

Cycling is a popular sport, and evaluation of the validity of tests to predict performance in competitions is important for athletes and coaches. Similarity between performance in sprints in mass-start bike races and in the laboratory is found, but, to our knowledge, no studies have investigated the relationship between laboratory measurements of maximal oxygen uptake (VO_2max_) and functional threshold power (FTP) with performance in official mass-start competitions. The purpose of this study was to evaluate the validity of a 20 min FTP test and VO_2max_ as predictors for performance in an official mountain bike competition. Eleven moderately trained male cyclists at a local level participated in this study (age: 43 ± 5.1 years; height: 183.4 ± 5.4 m; weight: 84.4 ± 8.7 kg; body mass index: 25.1 ± 2.1). All subjects performed a 20 min FTP test in the laboratory to measure the mean power. In addition, the subjects completed an incremental test to exhaustion to determine VO_2max_. These two laboratory tests were analyzed together with the results from a 47 km mass-start mountain bike race, with a total elevation of 851 m. A significant relationship was found between the mean relative power (W/kg) for the 20 min FTP test and performance time in the race (*r* = −0.74, *P* < 0.01). No significant correlation was found between VO_2max_ and cycling performance for these subjects (*r* = −0.37). These findings indicate that a 20 min FTP test is a more valid test for prediction of performance in mass-start bike races than a VO_2max_ test for moderately trained cyclists.

## 1. Introduction

Cyclists are characterized by their ability to produce high power during competitions lasting from 30 min to 6 h [1]. In an attempt to control and evaluate the training progress, testing of physiological and performance-related variables is essential [1]. The considerable variability of cycling tests makes it difficult for athletes and coaches to choose the most appropriate one for predicting competition performance [2].

Maximal oxygen uptake (VO_2max_) is probably the predictor that is tested the most for athletes in endurance-based sports. Testing of VO_2max_ is common because high oxygen uptake is an established indicator of aerobic endurance [3]. The use of changes in VO_2max_ as a marker for improvement in cycling performance is somewhat argued in the literature [2]. A high correlation between VO_2max_ and functional threshold power (FTP, i.e., the highest power a cyclist can maintain in a quasi-steady state without fatiguing for approximately 1 h [4]) was found when testing untrained recreational cyclists and moderately trained cyclists [5]. Similar correlations were found when investigating the relationship between peak power output (PPO), VO₂_max_, and performance time in a 20 km time trial (TT) for trained cyclists [6]. However, no significant association was found between VO_2max_ and cycling time in a half Ironman (90 km bike race) or a full Ironman (180 km bike race) [7]. It was discovered that, despite no differences in VO₂_max_ between groups, the “elite-national class” group performed a 40 km TT 10% faster than the “good-state class” group [8]. Research reported that VO₂_max_ did not correlate with performance in a 20 or 90 min TT for highly trained cyclists [9]. In contrast to this, another study found moderately high and high correlations (*r* = 0.61, 0.79 and 0.87 for 5, 20, and 60 min, respectively) when examining the relationship between VO_2max_ and performance in moderately trained cyclists [10]. Research indicates that VO₂_max_ is important for cycling performance at a moderate level, but, for high-performance athletes, there might be criteria for an optimal level of VO₂_max_ beyond which any further increase does not lead to increased cycling performance [5]. An explanation for this notion could be that the difference in VO₂_max_ for different performance levels is small, with values reported to be 70–75, 68–75, and 65–73 ml/kg/min among professional, elite, and amateur cyclists, respectively [2].

Cyclists depend on their ability to produce high levels of power. A high correlation is observed between absolute PPO measured during laboratory tests and power on the flat in a field TT [11,12,13,14]. Quod [15] found that power produced in a laboratory sprint test correlated highly with power in sprints during a mass-start competition. Absolute PPO is associated with performance time in a 20 min TT (mainly flat terrain) for well-trained cyclists [11], but this correlation is not found for cyclists at a moderate level [12]. During a 60 min laboratory TT, also called an FTP test, cyclists are instructed to generate as much power as possible for the entire hour. A shorter 20 min TT was found to be highly associated with performance in the 60 min TT [4]. Furthermore, large correlations (*r* = 0.61–0.88) were found between a 20 min FTP test (FTP20) and a 60 min FTP test (FTP60) on power output, heart rate, and oxygen uptake. The small difference between mean power during the FTP20 and FTP60 is argued to be due to a larger contribution of anaerobic metabolism in the FTP20 (for a review, see [16]); therefore, calculation of FTP is recommended by subtracting 5% of mean power in the FTP20 to find FTP [16].

The body mass of cyclists is important in establishing the ability to perform over different terrains in road racing. Bigger cyclists have an advantage on flat terrain, whereas small cyclists perform better cycling up hills [17]. The highest energy cost over flat terrain is due to air resistance, and larger cyclists with higher absolute power achieve relatively lower air resistance than smaller cyclists [17]. When cycling up hills, overcoming gravity causes the highest energy cost, and air resistance is less because of the lower speed [17]. Tan and Aziz [18] suggest that absolute power is a good predictor of cycling performance on flat terrain, whereas high relative power is a better predictor for cycling performance up hills. Other studies report relative PPO and VO_2max_ to be the best predictors for uphill cycling performance [19,20]. In mountain bike races, the importance of high levels of VO_2max_ and PPO related to body mass was highlighted by several studies [21,22,23]. However, one study reported that for high-level cross-country off-road cyclists, submaximal levels of aerobic fitness at the respiratory compensation point were more decisive for performance than PPO and VO_2max_ [24]. Performance in mass-start bike races not only depends on physiological abilities but also on tactical decisions, among others. The advantage of drafting increases with speed, and riding 40 km/h at the back of eight cyclists reduces VO_2max_ by 39% [25].

Cycling is a popular sport, and evaluation of the validity of tests to predict performance in competitions is important for athletes and coaches. Similarity between performance in sprints in mass-start bike races and in the laboratory was reported [15], but, to our knowledge, no studies have investigated the relationship between laboratory measurements of VO_2max_ and FTP with performance in mass-start competitions. The high correlation between an outdoor TT and indoor measurement of PPO could indicate that an indoor FTP test will correlate with performance in competitions [11,12,13,14]. This correlation between PPO and performance time in a TT was reported in two studies [6,19], both in an uphill TT and a mainly flat TT. In contrast, Balmer, Davison, and Bird [12] found no correlation between the same variables. The use of a power meter (to measure the power produced by cycling or spinning a bike) is popular and widespread among cyclists, making it easy for them to perform an FTP test in their own training area. Therefore, the purpose of this study was to investigate the validity of VO_2max_ and power measurements in an FTP test (absolute and relative power) compared to performance in a mass-start mountain bike competition. It was hypothesized that relative power would be the best predictor for performance in a mountain bike competition.

## 2. Material and Methods

### 2.1. Participants

Eleven moderately trained male competitive cyclists (age: 43 ± 5.1 years; height: 183.4 ± 5.4 m; weight: 84.4 ± 8.7 kg; body mass index (BMI): 25.1 ± 2.1) were recruited from a regional club to participate in the study. The inclusion criteria were a minimum of 2 years of experience in regional cycle competitions and between 200–500 training hours throughout the year. Following an explanation of the procedures, all participants gave written informed consent to participate in the study. The study was conducted according to the Declaration of Helsinki (2013) and approved by the Norwegian centre for Research Data.

### 2.2. Testing Procedures

All participants completed one laboratory testing session and one official mass-start mountain bike competition. In the laboratory test situation, participants performed a structured laboratory protocol that included a standardized 15 min warm-up. After the warm-up, cyclists performed a 20 min TT to determine the FTP [1]. Allen and Coggan [1] have suggested that the FTP60 could be determined as 95% of the mean power output in a 20 min TT, and this test was chosen because it is less time-consuming. Therefore, the FTP data in this study are 95% of the mean power performed in a 20 min TT. After 5 min of recovery, an incremental test to exhaustion was performed to determine VO_2max_. This test started with 1 min of cycling at 20 W below the average FTP20 value. Workload was subsequently increased by 25 W every minute until voluntary exhaustion or lack of further increase in O_2_ uptake. Verbal encouragement was provided throughout the test. VO_2max_ was defined as the average of the two highest VO_2_ measurements [26]. All participants were familiar with the test situation and completed a minimum of one laboratory test session in which they performed exactly the same tests before the final laboratory test.

The official race was a mass-start 47 km mountain bike race, where the first 22 km contained several uphills and the rest was mainly flat or downhill (see Figure 1 for race profile). Total elevation was 851 m. In the official competition, an electronic timing system (emit) was used. Performance in this mass-start 47 km bike race was tested by a system that included a radiofrequency identification timing tag attached to the participants. A radiofrequency identification reader/controller with antenna assembly (EQ Timing) captured the participants’ race time.

### 2.3. Instruments

All tests were conducted under standardized laboratory conditions of 20–22 °C and 50–60% relative humidity. Tests were performed on a Tacx Fortius cycle ergometer (Tacx B.V., Wassenaar, Netherlands), which is a stationary power meter that can be used in laboratory conditions. The unit consists of a brushless motor attached to a drum that, when in contact with the rear tire of an individual’s bicycle, creates resistance by acting as a generator/dynamo, converting power to alternating current. Before each test, the ergometer was calibrated according to the manufacturer’s recommendations. This equipment was scientifically validated and seemingly overestimates power output, but it is stable, and the measurements are reproducible [27]. The saddle height and distance between the tip of the saddle and the bottom bracket were adjusted by each participant as desired. During all laboratory tests, the participants were allowed to choose their preferred cadence.

VO_2_ was measured using Oxycon Pro (Oxycon: Jaeger GmbH, Hoechberg, Germany) with a mixing chamber and a 30 s sampling time. Gas sensors were calibrated via an automated process using certified calibration gases of known concentrations (15% O_2_, 6% CO_2_) and atmosphere air before every test. The flow turbine (Triple V, Erich Jaeger, Hoechberg, Germany) was automatically volume-calibrated according to the manufacturer’s recommendations.

### 2.4. Statistical Analysis

Data are expressed as mean ± standard deviation (SD) or as individual values. The distribution of each variable was examined for the assumption of normality using the Kolmogorov–Smirnov test. Correlations were determined using Pearson’s product moment correlation coefficient (*r*). The magnitudes of the correlation coefficients were stratified into groups comprising trivial (*r* < 0.1), small (0.1 < *r* < 0.3), moderate (0.3 < *r* < 0.5), large (0.5 < *r* < 0.7), very large (0.7 < *r* < 0.9), nearly perfect (*r* > 0.9), and perfect (*r* = 1.0) [28,29]. The entire statistical analysis was performed using SPSS Statistical Analysis Software for Windows^®^ (SPSS, version 25, Chicago, IL, USA).

## 3. Results

This study found a significant association between mean relative FTP and official race time (*r* = 0.74, *p* = 0.01) (see Figure 2A and Table 1). No significant correlation between any of the other variables was found: FTP (W), VO_2max_ (ml kg^−1^ min^−1^), weight (kg), and total time of race competition (min) (see Figure 2B–D and Table 1).

## 4. Discussion

The purpose of this study was to investigate the validity of VO_2max_ and power measurements in an FTP test (absolute and relative power) compared to performance in a mass-start mountain bike competition. Our main finding indicates a large correlation between FTP relative to body mass (W/kg) and performance in an official 47 km mass-start mountain bike race (r = −0.74, p < 0.01). Secondly, a high VO_2max_ was not associated with reduced time in the official 47 km mass-start mountain bike race for moderately trained cyclists.

Findings from this study show that results from an FTP test can give moderately trained cyclists important information about their cycling competition performance. High relative power on an indoor 20 min FTP test is associated with reduced cycling time in the official bike competition. In performance tests such as the FTP test, results depend on physiological ability, as well as work economy and psychological aspects concerning performance. A large correlation between laboratory measurements of absolute power at the lactate threshold (LT) or PPO and mean power in time trials was found in several studies [11,12,13,14]. Moreover, a significant relationship between PPO, (both relative and absolute power) and performance time in a mainly flat TT for well-trained cyclists was reported [6], as well as in an uphill TT [20]. For moderately trained cyclists with a mean age of around 40 years, the association between power output and performance time is more uncertain [12]. Lamberts (2011) found that a submaximal cycle test predicted performance time in a 40 km flat TT [30], whereas Balmer et al. (2000) found absolute peak power in a laboratory test to predict absolute power during an outdoor 16.1 km TT, but no significant correlation between PPO and performance time [12]. However, some weaknesses were presented in their study with regard to environment standardization and self-selected racing position. They investigated absolute peak power, but an examination of relative power could have given a higher correlation. In the present study, all subjects were able to adjust to a self-selected racing position, and the test situation indoors was standardized. However, in an official competition, many external variables (weather, temperature, environment, and race position) have an influential role in performance time. Laboratory tests often rely on expensive equipment, and affordability can be difficult for coaches and athletes. For high-level cyclists, cost or access to this equipment is more realistic than for moderately trained subjects. Therefore, our findings of very large correlations between relative power in the FTP test and cycling time in official competition could give valuable insight for coaches and athletes to monitor the training and performance for this cohort of athletes.

Another important difference between the present study and others investigating associations between power output and a TT is that during a TT, the rules forbid cyclists from drafting, which leads to an individual race against time. A mass start includes more tactical decisions that will possibly play an influential role in the performance. The participants in this study had an average relative FTP of 3.3 W/kg, which is categorized as a moderate level, while male world-class cyclists have a relative FTP of around 6, well-trained club cyclists have one between 3.5 and 4.5, and untrained cyclists have one around 2 [1]. This indicates that a good ability to produce high levels of power during competitions is a major determinant in bike races; this is supported by others who show that the difference between cyclists at a high level is not VO_2max_ but their ability to produce power [8].

In the present study, VO_2max_ seems of less importance in order to explain performance in a mass-start cycling competition. Contrary to our findings for moderately trained cyclists, both Borszcz et al. [4] and Denham et al. [5] found VO_2max_ to be a good predictor for performance in moderately trained cyclists. In Denham et al. [5], half of their participants were untrained. In contrast, the cyclists in our study trained 200–500 h throughout the year, mainly by cycling. In both studies, Denham et al. [5] and the present study, the relative VO_2max_ of the participants was 46 ml kg^−1^ min^−1^, tested by cycling. According to the specificity principle, cycling several hundred hours each year should improve cycling performance but does not necessarily have any impact on the VO_2max_ [31]. The race profile shows that the first 20 km of the race contained several uphills, while the last 26.7 km of the race was mostly flat and downhill, and the total elevation was 851 m. The participants in this study had a BMI of 25.1, which could indicate that they had some disadvantage in the first part of the race (mostly uphill) but had an advantage in the last part (mostly flat terrain or downhill) [17]. The best predictors for performance uphill are found by other researchers to be average power output and VO_2max_, both normalized to body mass [21,22,23]. Therefore, cyclists with a low BMI would benefit when cycling on uphill terrain [17]. Another possible explanation for the low correlation between VO_2max_ and performance time in the mass-start mountain bike race in this study is that performance in competitions is partly dependent on tactical decisions.

No significant relationship was found between absolute FTP and performance. This is not surprising in this type of race, which had a total elevation of 851 m. According to Tan and Aziz [18], high absolute power is important for performance on flat terrain, whereas high relative power is decisive when cycling uphill. Long uphill sections where cyclists must struggle against gravity will be beneficial for cyclists with low body mass [18]. We found no significant correlation between weight and performance. As highlighted before, relative FTP is the decisive factor related to performance, not absolute FTP or weight. If the course in this study had been flat, cyclists with high absolute power production during the FTP test would definitely have performed better in the competition [18].

## 5. Practical Implications and Future Research

The strength of the present study is that relative power produced in a 20 min FTP test is highly correlated to performance during a mass-start bike race among moderately trained cyclists. The FTP test used in the present study is easy to conduct, both at home and in a community gym, and gives cyclists of different levels a valid test that could help them evaluate and control their training progress. This study also indicates that expensive testing of VO_2max_ in a laboratory is not necessarily cost-effective for moderately trained cyclists.

Generalization of the findings in this study must be done with care, as the sample size of this study only included eleven male cyclists. Therefore, these results have to be interpreted in light of a modest sample size. Moreover, we must keep in mind that the participants were moderately trained, with an average age of 43 years and a BMI of 25.1, so our results cannot be generalized to all other groups of cyclists. In addition, performance in a mass-start bike race is not only dependent on physiological issues but also, for example, on tactical ability and psychological factors [25]. However, even though performance time is affected by many factors, high relative power in an FTP test is related to shorter cycling time in this type of cycling competition for this group. During the laboratory test, cyclists had only a 5 min recovery period after the 20 min FTP test before taking an incremental test to exhaustion to determine VO_2max_. It is possible that the levels of VO_2max_ could have been higher if this test had been taken the next day, but the ranking between cyclists would probably have been equal.

Future research should evaluate this validity for other types of cycling competitions and for cyclists at different levels and age groups. Furthermore, it would be interesting to evaluate the validity between relative power in an FTP test and cycling performance on flat terrain compared to uphills.

## 6. Conclusions

The present study demonstrate that relative power produced in a 20 min FTP test is highly correlated to performance during a mass-start bike race. However, no significant association was found between either absolute power or relative VO_2max_ and performance in a mass-start mountain bike race. 

## Figures and Tables

**Figure 1 sports-07-00217-f001:**
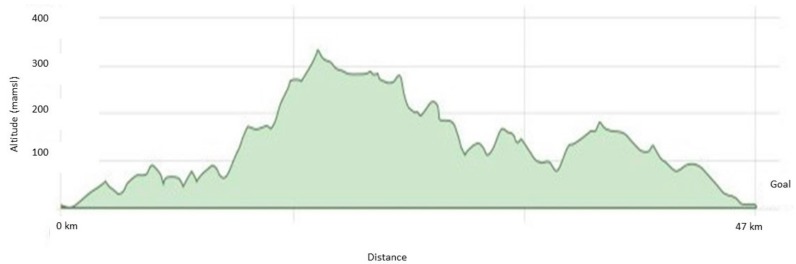
Race profile for the mass-start mountain bike race. Total distance: 47 km. Total elevation: 851 m. Altitude: 3–338 meters above mean sea level (MAMSL).

**Figure 2 sports-07-00217-f002:**
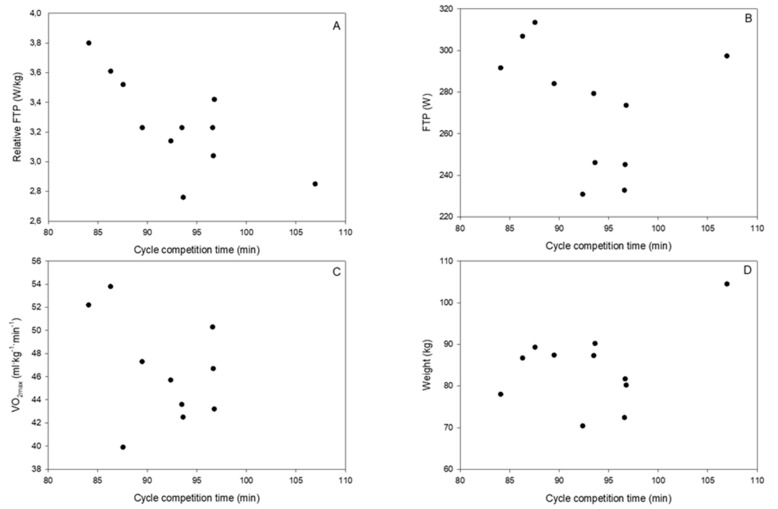
Plots of single sets of XY pairs for (**A**) relative FTP and race time; (**B**) FTP and race time; (**C**) VO_2max_ and race time; and (**D**) weight and race time, respectively. Each circle indicates one participant.

**Table 1 sports-07-00217-t001:** Pearson’s correlation (*r*) between performance variables (functional threshold power (FTP), FTP/kg, maximal oxygen uptake (VO_2max_), and race time) in eleven moderately trained competitive cyclists.

Physical Performance Variable	FTP	FTP/kg	VO_2max_	Race Time
FTP (*r*)	1	0.51	0.05	−0.29
FTP/kg (*r*)	0.51	1	0.45	−0.74 *
VO_2max_ (*r*)	0.05	0.45	1	−0.37
Race time (*r*)	−0.29	−0.74 *	−0.37	1
Total mean ± SD	272.8 ± 29.6	3.3 ± 0.3	46.3 ± 4.5	93.1 ± 6.3

* Correlation is significant at *p* < 0.01 (two-tailed).

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
