# Peer review of "The Validity of Functional Threshold Power and Maximal Oxygen Uptake for Cycling Performance in Moderately Trained Cyclists"

_sports, 2019, doi:10.3390/sports7100217_

Round 1

Reviewer 1 Report

The study evaluates the validity of a 20-min functional threshold power (FTP) test and VO2max as predictors for performance in an official mountain bike competition. The authors demonstrated that a 20-min FTP test is a more valid test for prediction of performance in mass-start bike races than a VO2max test for moderately trained cyclists.

The results of this research are in my opinion of relevance to the field of sports science and would fit the scope of the journal. There are, however, minor concerns which should be addressed.

Correlation coefficient usually indicates the degree of association but not of agreement. Please, consider to provide the Bland-Altman plot.

Page 4-5, lines 113-115: Therefore, the purpose of this study is to investigate the validity of power measurements in an FTP test (FTP20) compared to performance in a mass-start mountain bike competition.

Page 9, lines 207-209: The purpose of this study was to evaluate the correlations between physiological values of a 20-min FTP test and VO2max measured in a laboratory with performance in a mass-start mountain bike race.

Please, set the hypothesis / hypotheses.

It is not known whether the sampling was sufficient. It seems that the power analysis for estimating appropriate sample size was not provided.

Page 5, lines 119-122: In order to test the validity of the FTP test and VO2max as predictors of race 119 competition level, we tested 11 moderately trained male competitive cyclists (age: 120 43 ± 5.1 years; height: 183.4 ± 5.4 m; weight: 84.4 ± 8.7 kg; body mass index [BMI]: 121 25.1 ± 2.1).

Please, specify inclusion criteria for subjects to be allocated to the study.

Page 5, lines 122-123: All subjects had experience in regional cycle competitions and had 200-500 training hours throughout the year.

Reviewer 2 Report

 The validity of functional threshold power and maximal oxygen uptake for cycling performance in moderately trained cyclists

Line 8: what kind of cycling? Road cycling?

Line 16: age group athletes? Recreational athletes?

Lines 32-33: add a reference

Lines 33-35: add a reference

Lines 35-37: add a reference

Line 82: mountain bike race?

Line 109: found no correlation

Line 115: what is the hypothesis of your study?

Line 118: how were the subjects recruited? What were the criteria for inclusion/exclusion?

Lines 129 and 130: race and competition are the same term

Line 147: race and competition are the same term

Line 220: A high correlation

Line 308: add strength, weakness, implications for future research and practical applications for athletes and coaches
